# Design and Ballistic Performance of Hybrid Plates Manufactured from Aramid Composites for Developing Multilayered Armor Systems

**DOI:** 10.3390/polym14225026

**Published:** 2022-11-19

**Authors:** Cheng-Hung Shih, Jhu-Lin You, Yung-Lung Lee, An-Yu Cheng, Chang-Pin Chang, Yih-Ming Liu, Ming-Der Ger

**Affiliations:** 1Graduate School of Defense Science, Chung Cheng Institute of Technology, National Defense University, Taoyuan 335, Taiwan; 2Department of Chemical & Materials Engineering, Chung Cheng Institute of Technology, National Defense University, Taoyuan 335, Taiwan; 3System Engineering and Technology Program, National Yang Ming Chiao Tung University, Hsinchu 300, Taiwan; 4Department of Power Vehicle and Systems Engineering, Chung Cheng Institute of Technology, National Defense University, Taoyuan 335, Taiwan

**Keywords:** aramid composites, Impact resistance, energy absorption, ballistic affecting mechanism, failure modes, back face deformation

## Abstract

In this study, the impact resistance of aramid fabric reinforced with shear thickening fluids (STFs), epoxy or polyurea elastomers is examined through ballistic tests. According to the ballistic test results, the aramid composite structure treated with polyurea elastomers absorbs the most impact energy per unit area density and has the best impact resistance. However, the occurrence of stress concentration during ballistic impact reduces the impact resistance of the aramid composite structure treated with epoxy. On the other hand, aramid fabric impregnated with STF improves structural protection, but it also increases the weight of the composite structure and reduces the specific energy absorption (SEA). The results of this study analyze the energy absorption properties, deformation characteristics, and damage modes of different aramid composites, which will be of interest to future researchers developing next-generation protective equipment.

## 1. Introduction

The development of multilayered armor systems has evolved from hard metal materials to soft and high-performance fiber materials to meet the needs of users to wearing comfortable and lightweight equipment [1,2,3,4]. To enhance the performance of high-performance fibers used in soft body armor, many researchers currently use the concept of composite materials to produce fiber composites with properties superior to those of conventional high-performance fibers [5,6,7]. Composite materials have the advantages of being lightweight, having excellent corrosion resistance, high strength, high rigidity, and high design freedom, thus they have been widely used in a variety of products and fields such as aircraft, ships, automobiles, sports equipment, industrial production equipment, and civil engineering [8,9]. In the development of high-performance fiber composite materials for impact resistance, scientists have tried different methods to develop high-performance fiber composite materials with high strength, high modulus, and good compression resistance [10,11,12], which makes bulletproof vests lighter and more comfortable.

The new soft armor materials that have been currently studied by many researchers mainly use high-performance fibers combined with various thermosetting or thermoplastic polymers to form better fabric composites [13,14,15]. Under the impact of a projectile, the woven structure fabrics resist impact first, and then the impact force is transferred to other fibers by the two-dimensional (2D) woven structure. With polymer coating, the impact energy can be diffused more effectively and the impact energy can be dissipated [16,17,18]. Consequently, it enhances impact resistance. In addition, the new concept of liquid armor combines Kevlar fabric with impregnated shear thickening fluids (STFs), which successfully improve the impact resistance of Kevlar [19]. These STF/Kevlar composites dissipate the impact energy by the shear thickening properties of STF and improve the impact resistance [20,21,22,23,24,25]. The use of STF does improve the impact resistance of Kevlar, but it also increases the weight [26], which is not conducive to the development of lighter personal protective equipment.

The aramid fabric (Kevlar) was used as the main part of the hybrid plates in this study. Compared with other synthetic fibers, aramid fibers have unique advantages [27,28,29,30] and provide excellent protection against projectile impact [31]. Kevlar fibers have excellent properties, such as low density, high toughness, high strength, high-temperature resistance, and chemical corrosion resistance, and are easy to machine and mold [32,33]. Factors such as warp and weft yarn density, tissue structure, and interlacing points of single-layer fabrics will affect the ballistic resistance. According to the bulletproof mechanism and energy absorption mechanism of plain fabric, the interlacing point in the single-layer fabric will cause the reflected stress wave to be generated when the projectile makes impact, and then the compression point in contact with the projectile will be subjected to longitudinal force, lateral force and shear force at the same time. In order to further improve the impact resistance performance, but there are many experimental and theoretical results in the literature showing that the introduction of impurities into the fabric can not only improve the friction between the yarns but also improve the impact resistance [34,35,36]. The addition of impurities (filler particles) to the resin matrix can enhance toughness and energy absorption properties due to the energy mechanism absorption during crack propagation. In a study by Kim et al. [37], a simply sprayed polymer was proposed as a possible way to enhance the performance of ballistic fabrics in comparison to high-performance fibres impregnated with STF, and it was shown that the change in the movement of the polymer between the fibres and the STF increased the friction between the fibres, thus enhancing the protection. However, the structure of the fabric plays a crucial role in determining its impact resistance during high-speed impact [38]. Therefore, it is necessary to use appropriate adding materials and processing methods in order to produce a structure that has ballistic properties to avoid the weakness of ballistic materials caused by the phenomenon of stress concentration during projectile impact [39,40,41].

In the development and design of lightweight protective materials, mass effect is a major factor affecting the protective performance. Therefore, we try to use Kevlar fabric combined with STF, thermosetting epoxy resin, and polyurea elastomers to develop lighter and thinner multilayered armor systems. The dynamic mechanical and mechanical properties of each aramid composite structure were analyzed by ballistic testing. In the experiment, the scanning electron microscope was used to observe the microscopic destruction of fibers in the final static state and to understand the failure mode of each aramid composite structure. In addition, a high-speed camera was used to record the backside deformation of different types of targets due to different energy dissipation/absorption mechanisms during the projectile impact. This is another method that mainly focuses on the dynamic mechanical properties from experimental work to investigate the aramid composites impact responses and different failure mechanisms.

## 2. Experimental

### 2.1. Preparation of Hybrid Plates

In this study, Kevlar fabric (DuPont Protection Technologies, Richmond, WV, USA) with a size of 13 cm × 13 cm was used as the target plate. The processing and preparation methods of various aramid composites are described in the following sections:

#### 2.1.1. Preparation of STF/Kevlar Plates

Since the solid content of STF will affect the impact resistance, the experimental design of this study was based on the results reported by A. Khodadadi et al. [42]. The STF was composed of 35 wt% nanosized (10–30 nm) silica particles (ECHO Chemical Co., Ltd., Miaoli, Taiwan.) and 65 wt% polyethylene glycol (PEG, ECHO Chemical Co., Ltd., Miaoli, Taiwan.) in this study. The average molecular weight of the PEG is 200 g/mole. A planetary mixer (MJ-4000V, CGT Technology Co., Ltd., Taipei, Taiwan.) was used to mix nanosilica with polyethylene glycol for STF preparation. In the impregnating process, the STF is diluted with ethanol at a ratio of 1:1 by weight, and then the Kevlar fabric is impregnated with the diluted solution so that the STF penetrates into the Kevlar fabric. After that, the excess STF from the STF/Kevlar plate was removed and the fabrics were then dried at 70 °C for 8 h in an oven to evaporate the ethanol from the fabrics. Finally, the STF/Kevlar plates are assembled according to the number of layers needed in the ballistic test.

#### 2.1.2. Preparation of Epoxy/Kevlar Plates

The epoxy resin used in this section is a thermosetting resin provided by YizTech Co., Ltd., Taoyuan, Taiwan and the main components are Polyoxpropylenedimine and Diglycidyl Ether of Bisphenol F. Due to the long curing time and high flowability of acrylic resin at room temperature, the Epoxy/Kevlar plates were prepared by impregnation. After that, we prepared the Epoxy/Kevlar plates according to the number of layers of the ballistic test sample.

#### 2.1.3. Preparation of Polyurea Elastomers/Kevlar Plates

The polyurea elastomers used in the study were polymerized with aromatic diphenylmethane isocyanate (A agent) and a terminal amino polyether resin (B agent) provided by YizTech Co., Ltd., Taoyuan, Taiwan. Agent A and B were diluted with tetrahydrofuran (THF) in the ratio of 1:5 by weight at room temperature, and then the diluted solution was mixed and brushed onto Kevlar fabric to produce the Polyurea elastomers/Kevlar composites. Finally, Polyurea elastomers/Kevlar plates are prepared according to the number of layers discussed in the ballistic test. Polyurea is a highly reactive, non-polluting, and solvent-free spray coating that has been developed in recent years and is widely used because of its excellent adhesion, high strength, fast curing rate during synthesis, and insensitivity to ambient temperature and humidity.

### 2.2. Scanning Electron Microscopy (SEM) Analysis

The impact load transferred to the aramid composites by the penetration process of the projectile is mainly a concentrated load, which will cause serious deformation of the aramid composite structure in the impact zone, further causing the fracture and failure of the fiber structure [43]. Scanning electron microscopy (SEM; JSM-IT100, JEOL, Ltd, Singapore.) was used in this study to observe the aramid composites, and their microscopic fiber fracture after ballistic testing will be discussed and analyzed in comparison with the case of neat Kevlar fabric.

### 2.3. Ballistic Impact Testing

The experimental test method in this section can be divided into two parts: the projectile penetrating target and the non-penetrating target. The testing method for the penetration of the projectile through the target is based on the law of conservation of energy, which is used to calculate the energy absorption of the different aramid composites used in this study. During the test, the sample was fixed with a square aluminum plate with an opening area of 10 cm × 10 cm. When the projectile strikes the fabric of the woven fabric, a “wedge through” deformation pattern is generated [44]. Therefore, a high-speed camera (FASTCAM SA1.1, Tech Imaging Services, Inc., Saugus, MA, USA) was utilized to observe the dynamic back face deformation of different aramid composites when they are impacted by a projectile in order to further understand the ballistic affecting mechanism. The ballistic tests are conducted according to the National Institute of Justice NIJ-standard 0101.06 Type II and Type IIIA [45]. Ballistic testing was performed using 9 mm Full Metal Jacketed Round Nose (FMJ RN) bullets and 0.44 Magnum Semi Jacketed Hollow Point (SJHP) bullets. In fact, the velocity of the projectile may be easily affected by the current environment, such as factors like temperature, humidity, the storage period of projectiles, etc. The initial velocities of 9 mm FMJ RN bullets and 0.44 SJHP bullets are 412 ± 6 m/s and 442 ± 18 m/s. The appearance of the two types of projectiles under study are shown in Figure 1a,b, and the schematic diagram of ballistic testing equipment is shown in Figure 1c.

## 3. Results and Discussion

### 3.1. Surface Morphology of Aramid Composites

This work attempts to enhance the impact resistance of Kevlar fabric by using the concept of composite materials. By adding reinforcing materials, we try to stabilize the structure of Kevlar fabric and improve the impact responses to enhance the protection. Figure 2 shows the microscopic surface morphology of each aramid composite after using SEM magnification at the magnification of 50 times, and the magnification of 500 times. From Figure 2a,b, it can be observed that the neat Kevlar fabric is a 2D woven structure where the diameter of the single yarn is about 10 μm. Figure 2c,d show the surface appearance of Kevlar fabric impregnated with STF. It can be observed that the impregnated STF is distributed in the gaps between the yarns, but there are still some gaps between the warp and weft yarns that are not filled with STF. In Figure 2e,f, it can be seen that the Kevlar fabric is completely covered by epoxy and the gaps between the yarns are filled by the excess epoxy. Kevlar fabric and epoxy are combined to form a new 3D structure. Figure 2g,h show the surface topography of Kevlar fabric covered with Polyurea Elastomer. Similarly, it can be observed that the Kevlar fiber is covered by Polyurea Elastomers; however, there are still some interstitial spaces between the yarns in the fabric structure which are not completely filled up by the polyurea. This might be attributed to the fact that the coating of polyurea on the Kevlar fabric was performed with a diluted solution, resulting in the polyurea not being able to completely fill the interstices of the fabric structure.

### 3.2. Effect of Aramid Composites on Impact Energy Absorption

In this section, we use the 9 mm ammunition to conduct ballistic testing on single-ply aramid composites. Under ideal conditions, if the actual energy will be converted into heat and sound dissipation and other dissipation factors are not considered, the impact energy absorption by the test target can be calculated according to the law of conservation of energy, the calculation formula of which is shown in Equation (1) [46,47]. It was assumed that the energy absorption by the test target is equal to the energy lost by the projectile. In addition, in order to compare the energy absorption per unit area density of different targets and to understand the protective effect of different aramid composites prepared in the study after eliminating the mass influencing factors, Equation (2) is used to calculate the specific energy absorption (SEA) of different targets. The formula of calculation is as follows:(1)$$\Delta E_{a}=\frac{1}{2}{\mathrm{mV}}_{i}^{2}-\frac{1}{2}{\mathrm{mV}}_{r}^{2}$$
where ∆E_a_ is the energy absorption by the target to be measured (J), m is the mass of the projectile (kg), V_i_ is the initial velocity of the projectile (m/s), and V_r_ is the residual final velocity after the projectile penetrates the target (m/s).
(2)$$\mathrm{SEA}=\frac{\Delta E_{a}}{\mathrm{Areal}\;\mathrm{density}}$$
where SEA is the specific energy absorption (J·cm^2^/g), ∆E_a_ is the energy absorption (J) of the target obtained from the calculation of Equation (1), and Areal density is the areal density of different aramid composites (g/cm^2^).

The energy absorption and SEA results calculated from Equations (1) and (2) are shown in Figure 3. The trend of Figure 3 is the result obtained after 3 tests of different aramid composites. From the comparison of the results of energy absorption and SEA in Figure 3a, it can be seen that the impact energy absorption of aramid composites prepared in this study is Polyurea Elastomers/Kevlar plates > STF/Kevlar plates > neat Kevlar plates > Epoxy/Kevlar plates. In addition, the results of ballistic testing with 7-ply Polyurea Elastomers/Kevlar plates and 7-ply neat Kevlar plates are shown in Figure 3b. From Figure 3b, it can be seen that the improved 7-ply Polyurea Elastomers/Kevlar plates increase the total energy absorption by 158% and the SEA can be increased by 153% compared to the 7-ply neat Kevlar plates control group. This result also shows that the Polyurea Elastomers/Kevlar composite material is more efficient in absorbing energy.

### 3.3. Damage in Fracture Analysis

The ballistic of aramid composites will be affected by some factors (such as mechanical properties and micro-geometry) during the high-speed impact of the projectile [9]. From the macroscopic topography, Figure 4a,b show the original appearance of the 1-layer ply of neat Kevlar fabric and its damage after ballistic testing. Figure 4b shows the neat Kevlar fabric after the penetration of the projectile, due to the relationship between the yarn pulling each other to produce the intermediate perforations and cross-shaped pulling marks. Similarly, damage patterns with similar perforations and cross-shaped pulling marks can be observed in Figure 5a,b and Figure 6a,b. The results show that the structures of STF/Kevlar and Polyurea Elastomers/Kevlar composites retain the mechanism of energy dissipation by friction between the yarns in the Kevlar fabric. Figure 7a,b display the original appearance and damage after ballistic testing for Epoxy/Kevlar plates. It can be clearly seen from Figure 7 that only the perforation damage is present, but no cross-shaped pulling marks are left. This shows that the projectile does not produce a large area of yarn movement and yarn stretching mechanism during the process of penetration of Epoxy/Kevlar plates, which is also the reason why it absorbs less impact energy than the neat Kevlar plates. The addition of epoxy to neat Kevlar fabric affects the ballistic properties of the composite structure and leads to a weakening of the structure due to stress concentration.

During the impact of the projectile, the failure modes generated by the contact of the projectile with the target consist of tensile and shear damage generated by the contact of the projectile with the fabric. In Figure 4c,d, the microscopic damage pattern of neat Kevlar fiber at the fracture point after the ballistic test, it can be observed that the neat Kevlar fiber exhibits a tip shape due to tensile deformation after the impact damage fracture. Similarly, the microscopic damage pattern at the fracture point of Epoxy/Kevlar fiber after ballistic testing is observed in Figure 7c,d, which shows the tip shape due to tensile deformation. The result may be that when the projectile comes into contact with the Epoxy/Kevlar fiber, the epoxy instead creates a lubricating effect, causing the projectile to penetrate directly through the Epoxy/Kevlar fiber, weakening the protective effect of the Epoxy/Kevlar fiber. However, from the SEM images of the STF/Kevlar plates in Figure 5c,d at the fractured fibers after ballistic testing, in addition to a few STF/Kevlar fibers with a tip shape, there is also a part of the yarn that shows a tearing jagged appearance due to the shear force. The result may be that when the projectile touches the STF/Kevlar fiber, the rigid nanoparticles dispersed in the STF relatively increase the friction between the projectile and the fiber, prompting part of the impact loads can diffuse to the transverse section of the penetration path to dissipate the dissipation energy. As the frictional force increases the shear breaking force on the STF/Kevlar fiber, a tearing jagged appearance is formed in part of the yarn due to the shear force. Similarly, in the microscopic damage patterns of Polyurea Elastomers/Kevlar plates at the fracture point after ballistic testing in Figure 6c,d, a more jagged shape of tearing of the yarn due to shear forces is observed. This result also shows that the composite structure formed by the addition of Polyurea Elastomers to Kevlar fabric can more efficiently disperse the longitudinal impact energy of the projectile to the transverse cross-section.

### 3.4. The Dynamic Back Face Deformation Response of Single-Ply and Multi-Ply Target Plates to High-Velocity Impact Loading

To design suitable all-around personal protective equipment, it is important to understand the dynamic behavior of ballistic materials during and after impact, which will help to further identify the ballistic-affecting mechanism of aramid composites. The ballistic affecting mechanism of different aramid composites is mainly through the friction between yarns and the dissipation/absorption of impact energy through different mechanisms during the maximum deformation stage [48,49]. Unlike the static observation in the final damage in Section 3.3, this section uses a high-speed camera to record the dynamic back face deformation of projectile penetration through various single-ply composite fabrics, as shown in Figure 8. The high-speed camera is set at 30,000 frames per second to capture the entire process of the ballistic test. From the comparison of the time points of the projectile penetration, the time of the projectile penetration through the 1-layer ply aramid composites is almost the same, but the dynamic back face deformation caused by the impact of the projectile on the aramid composites from the same time points is very different. Figure 8a–c shows the dynamic back face deformation of the penetration of 1-layer ply neat Kevlar fabric by a projectile. It can be seen that when the projectile is in contact with the neat Kevlar fabric, it will pull the main contact yarn to form a cross pattern, and when the projectile penetrates the neat Kevlar fabric, it will form a higher cone protrusion, as shown in Figure 8b. During the dynamic process of a projectile penetrating a neat Kevlar fiber, a general woven fabric ballistic mechanism can also be observed, including yarn movement, yarn stretching, and yarn breakage. Through these mechanisms, the 2D woven structure fabrics can achieve the effect of dissipating the impact energy. On the other hand, by observing the dynamic process of projectile penetration of STF/Kevlar and Polyurea Elastomers/Kevlar plates in Figure 8d–f and Figure 8j–l, it can be found that both composites form a cone when the projectile is penetrated. However, comparing the results of Figure 8i and Figure 8l, it can be seen that the cone formed by STF/Kevlar fiber is higher and the circular area at the bottom is smaller than the cone portion formed by Polyurea Elastomers/Kevlar plates. This result also shows that Polyurea Elastomers/Kevlar plates composite materials are more efficient in dissipating the impact energy during the impact process, so Polyurea Elastomers/Kevlar plates composite materials absorb more of the kinetic energy of the projectile during penetration. However, from the dynamic process of projectile penetration of the 1-layer ply Epoxy/Kevlar plate in Figure 8g–i, it can be observed that the projectile shows the appearance of direct perforation and brittle fracture when penetrating the Epoxy/Kevlar plates. This is another proof of the theory that the addition of epoxy will cause a concentration of stress on Kevlar fabric during the impact process, which will reduce the impact resistance of Epoxy/Kevlar plates.

Since polyurea can absorb a large amount of impact energy before failure, it has been used as a protective coating or as an interlayer material for structures and composite systems in military applications such as blast shock and bulletproofing [37,50,51,52,53,54]. However, experimental observations of the single-ply fabric system could not show the interaction between the fabrics in the multilayered armor systems, so it is necessary to confirm this property by studying the target of the multi-ply fabric under impact loading. In the experiment to test the impact energy absorption by the 7-layer ply neat Kevlar plates and the Polyurea Elastomers/Kevlar plates, the penetration process of the projectile was recorded by the same high-speed camera, as shown in Figure 9. A comparison of the dynamic morphology during penetration of Figure 9a–d and Figure 9e–h shows that when the projectile is penetrated, the neat Kevlar plates produce a more winding pattern compared to the Polyurea Elastomers/Kevlar plates, resulting in a higher protruding shape. However, after the projectile perforation, the cross-sectional morphological changes shown in Figure 9d,h show that the Polyurea Elastomers/Kevlar plates exhibit a more extensive and stable radial diffusion mode than the protruding shape of the neat Kevlar plates. This result also shows that the addition of Polyurea Elastomers to neat Kevlar plates changes the mode of energy transfer between the fibers and slows down the back face deformation (BFD). The Polyurea Elastomers help to reduce the failures of the fabric’s impact load improvement by strengthening the fabric structure and later increase and facilitate the absorption of energy by the fabrics. 

### 3.5. Ballistic Test Results

In this section, Polyurea Elastomers/Kevlar plates with the best impact resistance are used for ballistic testing by NIJ-standard 0101.06. Due to the ballistic test results, the 13 cm × 13 cm neat Kevlar plates need 19 layers to protect the 9 mm ammunition by the NIJ-standard 0101.06 of 44 mm depth of depression in oil-based modeling clay. The 19-ply 13 cm × 13 cm neat Kevlar plates sample was code-named P1 and used as a control for ballistic testing with 9 mm ammunition. In the ballistic testing of the 0.44 ammunition, due to its higher warhead weight and faster bullet movement, the impact energy generated is greater than 9 mm ammunition. After ballistic testing, 47 layers of neat Kevlar plates are required to achieve the NIJ-standard 0101.06 protective effect. Ballistic testing requires 47 layers of neat Kevlar plates to achieve compliance with the NIJ-standard 0101.06 Type IIIA. The 47-ply 13 cm × 13 cm neat Kevlar plates sample code is P2, which is used as a control for ballistic testing of 0.44 ammunition.

Table 1 shows the ballistic test results of Polyurea Elastomers/Kevlar plates. If Polyurea Elastomers/Kevlar plates are not bonded to each other, at least 15 layers of Polyurea Elastomers/Kevlar fiber are required to resist the impact of 9 mm ammunition, by the NIJ-standard 0101.06 Type II. The 0.44 ammunition portion of the protection requires 37 layers to meet the requirements of NIJ-standard 0101.06 Type IIIA. If Polyurea Elastomers are used directly to bond the gaps between the layers, the impact resistance of Polyurea Elastomers/Kevlar plates can be enhanced because of the overall Polyurea Elastomers/Kevlar plates bonding relationship. Comparing the S5 sample with its control P2 sample, the number of Kevlar layers is reduced by 10 and the weight is reduced by 14.08%. In addition, the low depth of depressions in the oil-based modeling clay also shows that Polyurea Elastomers/Kevlar plates have better impact resistance and meet the requirements of lighter and more comfortable personal protective equipment in the future.

## 4. Conclusions

With the different needs of the actual wearers of body armor, new body armor specifications are gradually developing in the direction of lightweight, mobility, and wearing comfort. High protection and comfort are the main goals of today’s researchers in developing new personal protective equipment materials. To effectively strengthen the impact resistance of Kevlar fabric, this study used a composite material approach to obtain suitable materials and processing methods to improve the impact energy absorption efficiency. In the study, the absorption energy and SEA of the single-ply Kevlar fabric combined with STF nanomaterial, thermosetting epoxy, and Polyurea Elastomers were compared by ballistic testing. The test results show that Polyurea Elastomers/Kevlar plates have the best impact resistance. The 7-ply Polyurea Elastomers/Kevlar plates absorbed 158% more impact energy than the 7-ply neat Kevlar plates, and the SEA increased by 153%. Ballistic testing according to the NIJ-standard 0101.06 shows that Polyurea Elastomers/Kevlar plates have 4 fewer layers and 11.76% less weight than neat Kevlar plates under Type II testing. Under Type IIIA testing, Polyurea Elastomers/Kevlar plates require 10 fewer layers and weigh 14.08% less than neat Kevlar plates. Kevlar fabric processing by Polyurea Elastomers can reduce the thickness and weight of bulletproof equipment, which can be used for the development of new multilayered armor systems in the future.

## Figures and Tables

**Figure 1 polymers-14-05026-f001:**
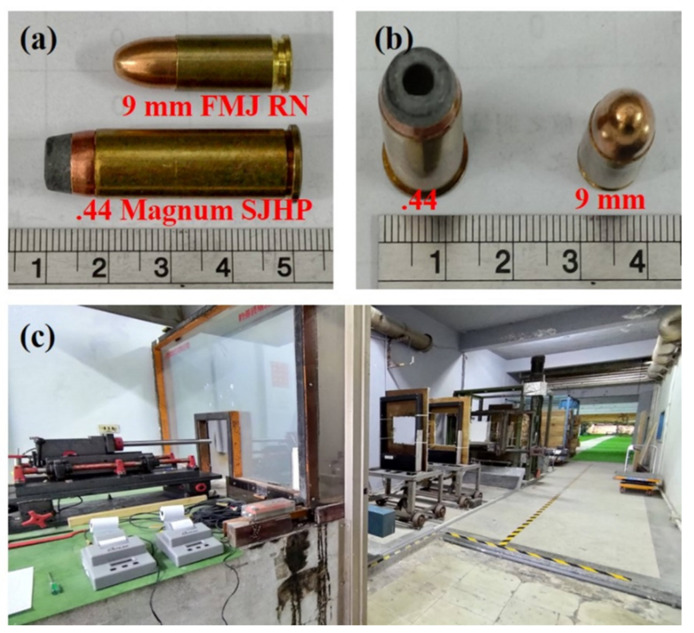
(**a**) Side view of 9 mm FMJ RN and 0.44 Magnum SJHP bullets; (**b**) Top view of 9 mm FMJ RN and 0.44 Magnum SJHP bullets; (**c**) Schematic set-up of the ballistic test setup.

**Figure 2 polymers-14-05026-f002:**
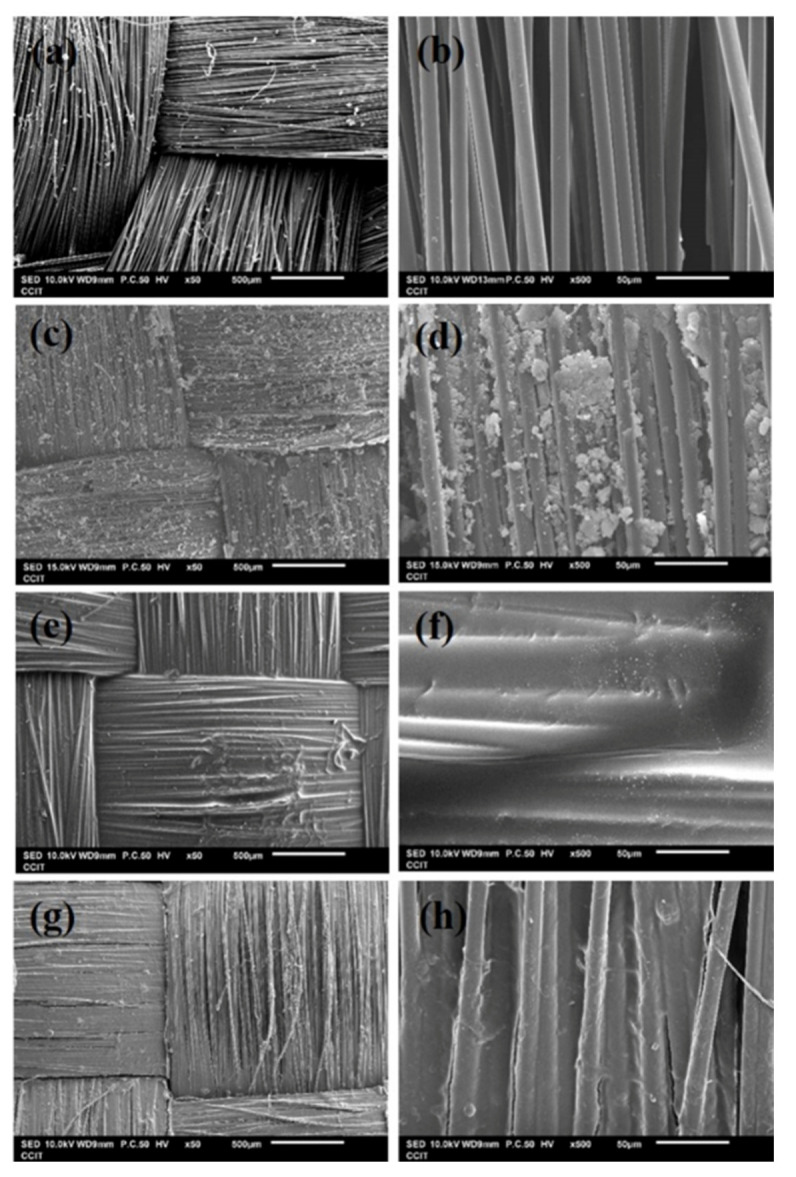
SEM images of the (**a**,**b**) Neat Kevlar plates; (**c**,**d**) STF/Kevlar plates; (**e**,**f**) Epoxy/Kevlar plates; (**g**,**h**) Polyurea Elastomers/Kevlar plates.

**Figure 3 polymers-14-05026-f003:**
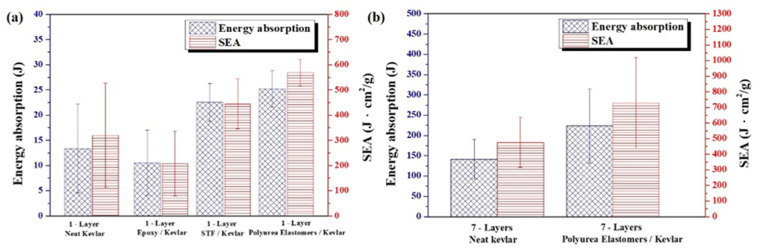
(**a**) Energy absorption and SEA of single-ply fabric; (**b**) Energy absorption and SEA of 7-ply fabrics.

**Figure 4 polymers-14-05026-f004:**
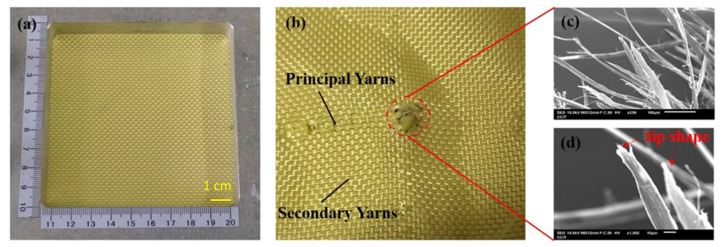
Observation of the single-ply (**a**) neat Kevlar plate before the ballistic test; (**b**) failed fabric with neat Kevlar plate after ballistic test; (**c**) the close-up look of the failed fibers in (**b**); (**d**) the zoomed-in image of the failed fibers in (**c**).

**Figure 5 polymers-14-05026-f005:**
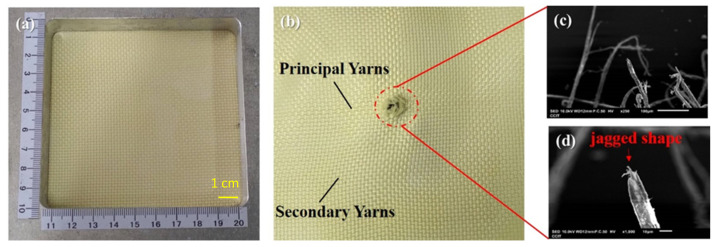
Observation of the single-ply (**a**) STF/Kevlar plate before the ballistic test; (**b**) failed fabric with STF/Kevlar fabrics plate ballistic test; (**c**) the close-up look of the failed fibres in (**b**); (**d**) the zoomed-in image of the failed fibers in (**c**).

**Figure 6 polymers-14-05026-f006:**
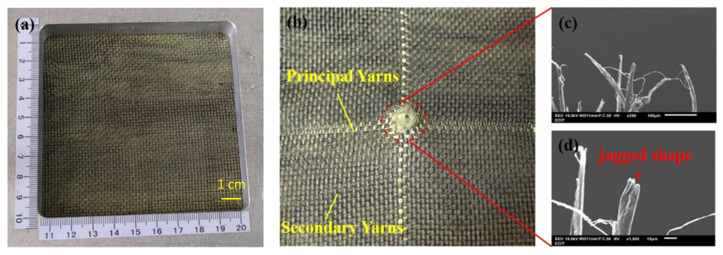
Observation of the single-ply (**a**) Polyurea Elastomers/Kevlar fabrics before the ballistic test; (**b**) failed fabric with Polyurea Elastomers/Kevlar fabrics after the ballistic test; (**c**) the close-up look of the failed fibers in (**b**); (**d**) the zoomed-in im-age of the failed fibers in (**c**).

**Figure 7 polymers-14-05026-f007:**
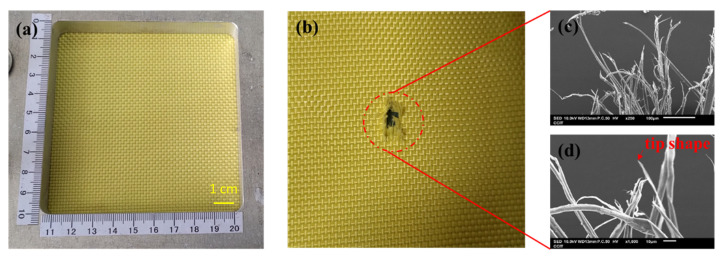
Observation of the single-ply (**a**) Epoxy/Kevlar plate before the ballistic test; (**b**) failed fabric with Epoxy/Kevlar plate after ballistic test; (**c**) the close-up look of the failed fibers in (**b**); (**d**) the zoomed-in image of the failed fibers in (**c**).

**Figure 8 polymers-14-05026-f008:**
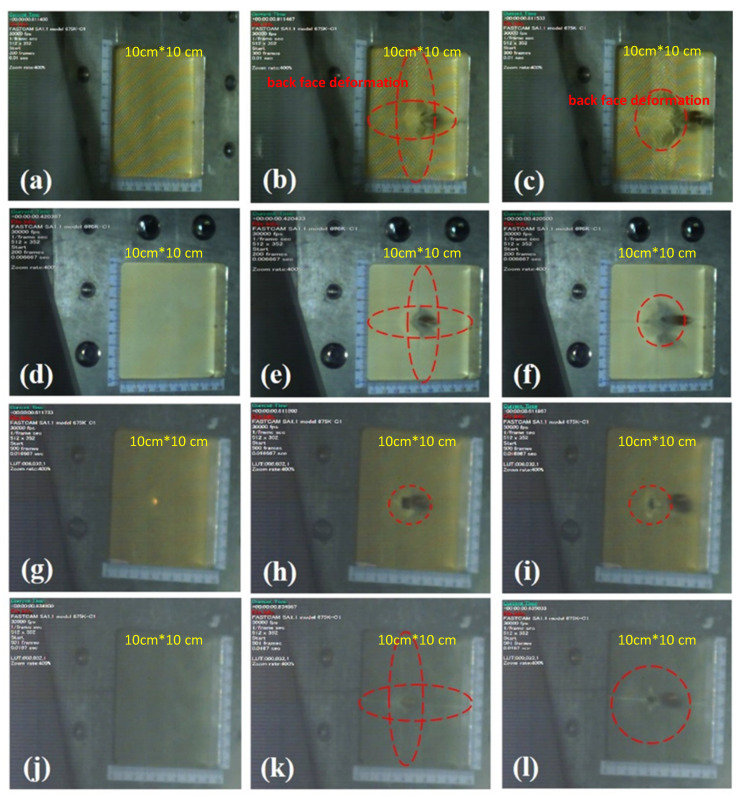
The dynamic back face deformation response of the single-ply (**a**) neat Kevlar plate at t = 0 s; (**b**) neat Kevlar plate at t = 6.7 × 10^−5^ s; (**c**) neat Kevlar plate at t = 1.3 × 10^−4^ s; (**d**) STF/Kevlar plate at t = 0 s; (**e**) STF/Kevlar plate at t = 6.7 × 10^−5^ s; (**f**) STF/Kevlar plate at t = 1.3 × 10^−4^ s; (**g**) Epoxy/Kevlar plate at t = 0 s; (**h**) Epoxy/Kevlar plate at t = 6.7 × 10^−5^ s; (**i**) Epoxy/Kevlar plate at t = 1.3 × 10^−4^ s; (**j**) Polyurea Elastomers/Kevlar plate at t = 0 s; (**k**) Polyurea Elastomers/Kevlar plate at t = 6.7 × 10^−5^ s; (**l**) Polyurea Elastomers/Kevlar plate at t = 1.3 × 10^−4^ s.

**Figure 9 polymers-14-05026-f009:**
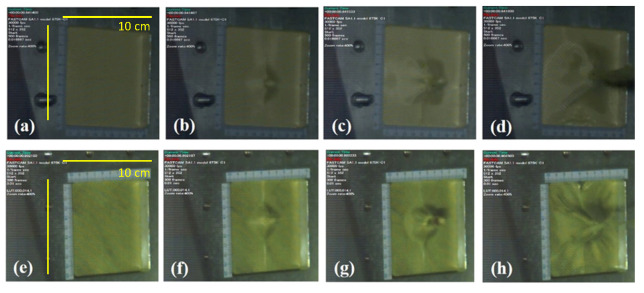
The dynamic back face deformation response of the 7-ply (**a**) neat Kevlar plate at t = 0 s; (**b**) neat Kevlar plate at t = 6.7 × 10^−5^ s; (**c**) neat Kevlar plate at t = 1.3 × 10^−4^ s; (**d**) neat Kevlar plate at t = 4.0 × 10^−4^ s; (**e**) Polyurea Elastomers/Kevlar plate at t = 0 s; (**f**) Polyurea Elastomers/Kevlar plate at t = 6.7 × 10^−5^ s; (**g**) Polyurea Elastomers/Kevlar plate at t = 1.3 × 10^−4^ s; (**h**) Polyurea Elastomers/Kevlar plate t = 4.0 × 10^−4^ s.

**Table 1 polymers-14-05026-t001:** The ballistic test results of Polyurea Elastomers/Kevlar plates with the NIJ-standard 0101.06 Type II and Type IIIA.

Results for Ballistic Test							
Plate ID	Composition	Layers	Areal Density (g/cm^2^)	Projectile Speed (m/s)	BFS (mm)	Ammo Type	NIJ 0101.06 Test Level
P1	Neat Kevlar	19	0.85	417	43.7	9 mm FMJ	II
S1	Polyurea Elastomers /Kevlar	17	0.83	413	36.2	9 mm FMJ	II
S2	Polyurea Elastomers /Kevlar	15	0.75	414	43.7	0.44 mag	II
P2	Neat Kevlar	47	2.06	447	43.1	0.44 mag	IIIA
S3	Polyurea Elastomers /Kevlar	24	1.10	451	66.3	0.44 mag	IIIA
S4	Polyurea Elastomers /Kevlar	37	1.76	446	42.6	0.44 mag	IIIA
S5	Polyurea Elastomers /Kevlar(solidify)	37	1.77	424	34.3	0.44 mag	IIIA

## Data Availability

The data presented in this study are available on request from the corresponding authors.

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
