# Peer review of "Design and Ballistic Performance of Hybrid Plates Manufactured from Aramid Composites for Developing Multilayered Armor Systems"

_polymers, 2022, doi:10.3390/polym14225026_

Round 1

Reviewer 1 Report

The authors deal with the problem of ballistic test of aramid panels reinforced with shear thickening fluids, epoxy or polyurea elastomer.

The authors should address some remarks:

1.       Section 2.1.1 describes generally the method of STF preparation and impregnating Kevlar. Please expand the description of the STF production method. What was the moisture content of the PEG200 used? Was there no degradation of the polyglycol matrix during drying (ethanol removal) as described in the manuscript (provide an article that is worth citing here (Materials2022, 15 (17), 5818;).

2.       The mechanical properties of a cross-linked epoxy resin depend on degree of cross-linking. Was the degree of cross-linking of the epoxy resin analysed? Please describe the manufacturing process in more detail (Epoxy / Kevlar plates and Polyurea Elastomers / Kevlar plates).

3.       In section 2.3, Ballistic impact testing, there is no information about the projectile’s velocity just before it hits the composite material. This is key information. Specific energy absorption (SEA) parameters depend on the projectile velocity.

4.       Please provide detailed information (for example in Table) on projectile velocities in individual samples for specific systems, i.e. epoxy and polyurea impregnated.

5.       Table 1: was there only one shot for each sample? If so, it is challenging to draw constructive conclusions from such studies.

6.       Table 1: does not show the results for the epoxy-cured samples.

Reviewer 2 Report

I have read the manuscript from the authors and I have to say that some points must be clarified. Moreover, the Authors should provide some references in some statements and correct some minor points. Please see pdf attached with all the comments.

Reviewer 3 Report

Minor Revision

1) Title: Ok

2)    Authors fellow the Polymer template

3)      Abstract:  English check is mandatory

4)      Keywords: you can add more keywords

5)      Add list of abbreviation and nomenclature

6)      Introduction: please add references literature studies of previous works on the same subject to show clearly the state of the art of your work.

7)      Which size you can consider your Kevlar fabrics plate ballistic is efficient.

8)      Results need more discussions emphasize the failed fibers why?

9)  References: fellow the Polymer template and update them by adding 2022 and 2023 

With regards

Round 2

Reviewer 1 Report

The authors have improved their work and included the reviewer's remarks. The authors improved the quality of the articles so that the paper can be accepted in its present form.